# Role of Mitochondria–ER Contact Sites in Mitophagy

**DOI:** 10.3390/biom13081198

**Published:** 2023-07-31

**Authors:** Alina Rühmkorf, Angelika Bettina Harbauer

**Affiliations:** 1TUM Medical Graduate Center, Technical University of Munich, 81675 Munich, Germany; 2Max Planck Institute for Biological Intelligence, 82152 Planegg-Martinsried, Germany; 3Institute of Neuronal Cell Biology, Technical University of Munich, 80802 Munich, Germany; 4Munich Cluster for Systems Neurology, 81377 Munich, Germany

**Keywords:** mitochondria, mitophagy, organellar contact sites

## Abstract

Mitochondria are often referred to as the “powerhouse” of the cell. However, this organelle has many more functions than simply satisfying the cells’ metabolic needs. Mitochondria are involved in calcium homeostasis and lipid metabolism, and they also regulate apoptotic processes. Many of these functions require contact with the ER, which is mediated by several tether proteins located on the respective organellar surfaces, enabling the formation of mitochondria–ER contact sites (MERCS). Upon damage, mitochondria produce reactive oxygen species (ROS) that can harm the surrounding cell. To circumvent toxicity and to maintain a functional pool of healthy organelles, damaged and excess mitochondria can be targeted for degradation via mitophagy, a form of selective autophagy. Defects in mitochondria–ER tethers and the accumulation of damaged mitochondria are found in several neurodegenerative diseases, including Parkinson’s disease and amyotrophic lateral sclerosis, which argues that the interplay between the two organelles is vital for neuronal health. This review provides an overview of the different mechanisms of mitochondrial quality control that are implicated with the different mitochondria–ER tether proteins, and also provides a novel perspective on how MERCS are involved in mediating mitophagy upon mitochondrial damage.

## 1. Mitochondrial Quality Control

Mitochondria are required for a multitude of cellular functions, including energy metabolism, calcium homeostasis, and lipid metabolism, as well as apoptotic and immune signaling [1,2,3]. Upon damage, mitochondria can enhance oxidative stress within the cell, leading to cell death [4,5]. Hence, maintaining a pool of healthy mitochondria is crucial for cellular survival, which is mediated by several mitochondrial quality control pathways. Defects in mitochondrial quality control pathways, and specifically mitophagy, are therefore linked to the pathogenesis of several neurodegenerative disorders including Parkinson’s disease and amyotrophic lateral sclerosis [6].

The first step of mitochondrial quality control is mediated by chaperones and proteases. Chaperones protect mitochondrial proteins from heat stress and promote the correct folding of preproteins entering the organelle in both the cytosol as well as inside the mitochondrial matrix. Proteases within the matrix and intermembrane space degrade irreversibly damaged or misfolded proteins to oligopeptides and amino acids [7]. The capacity of the chaperone and protease system within the mitochondria can be modified by the mitochondrial unfolded protein response (mtUPR). The mtUPR alters the nuclear expression of proteases and chaperones, and it increases mitochondria–ER coupling, supporting enhanced metabolic activity [8,9]. Misfolded or damaged proteins on the outer mitochondrial membrane (OMM) can be ubiquitinated and, subsequently, degraded by the cellular proteasomal system [10]. Interestingly, both the mitochondria and ER stress response rely on activating transcription factor 4 (ATF4) signaling [11], hinting at a concerted stress response from the two interconnected organelles.

However, if larger sections of the mitochondria are damaged, their effective removal via mitophagy, a specialized form of selective macroautophagy (hereafter referred to as autophagy) [12], is crucial for cellular homeostasis [13]. Both processes, mitophagy and autophagy, are not yet fully understood, and, while they share various similarities, some aspects may differ. These differences include the lipid source and composition of the autophagosomal and mitophagosomal membranes, respectively [14,15,16], as well as the autophagy adaptors and effectors required for the process [17,18]. The following paragraphs will provide an overview of the different mitophagy pathways.

Although mitophagy pathways can be very distinct in their orchestration, the pathways share similarities in their initiation and their termination [19]. Under homeostatic conditions, both fission and fusion play important roles in maintaining a functional mitochondrial network, and it is reported that alterations in mitochondrial dynamics precede and potentially even elicit mitophagy [20,21]. While mitochondrial fusion decreases upon mitophagy induction, mitochondrial fission increases, segregating damaged mitochondria from the healthy mitochondrial network, additionally resulting in smaller mitochondrial fragments [22]. This fragmentation facilitates the engulfment by mitophagosomal membranes [23]. Recently, the type of mitochondrial fission was described to differ between homeostatic- and damage-related events. While midzone fission leads to mitochondrial division at mitochondria–ER contact sites (MERCS) and results in two functional mitochondria, the peripheral type cleaves off the mitochondria with a reduced membrane potential and contacts lysosomes in advance [24]. While the first form has been suggested to serve the biogenesis of new mitochondria, the latter form leads to mitophagy.

Generally, autophagy can be divided in several sequential steps: initiation, phagophore elongation, autophagosome formation, fusion with the lysosome, and termination [25,26,27].

Autophagy is initiated via the phosphorylation of the general autophagy initiator kinases unc-51-like kinase 1 (ULK1) and ULK2. Phosphorylation is mediated by the key cellular sensors mechanistic target of rapamycin (mTOR) or adenosine monophosphate (AMP)-activated proteinkinase (AMPK) [28,29]. ULK1/2 then forms a complex with focal adhesion kinase (FAK) family kinase interacting protein of 200 kDa (FIP200), autophagy related (ATG) protein 13, and ATG101, referred to as the ULK1/2 complex [30,31].

The ULK1/2 complex in turn initiates the second step in autophagy, by phosphorylating and activating the class III phosphatidylinositol 3-kinase (PI3K) complex [32], consisting of activating molecule in Beclin 1-regulated autophagy (Ambra1), Beclin 1, Atg14L (also known as Barkor), vacuolar protein sorting 15 (VPS15), and VPS34 [33,34,35]. This complex promotes phosphatidylinositol 3-phosphate (PI3P) production, resulting in phagophore (also known as isolation membrane) growth. Further elongation of the membrane requires the ATG16 complex, consisting of Atg5, Atg12, and Atg16L1 [32]. The formation of the ATG16 complex is mediated by WD repeat domain phosphoinositide-interacting protein 1 (WIPI1), WIPI2, and double FYVE-containing protein 1 (DFCP1), which are recruited to the membrane by PI3P [36,37,38]. In parallel, Atg4 processes microtubule-associated protein 1A/1B-light chain 3 (LC3) so it exposes a glycine residue at its C-terminal (LC3-I) [39]. The Atg16 complex recruits the LC3 lipidation machinery, consisting of the E1 ligase Atg7 and Atg3 [40], which covalently links LC3-I in an ATP-dependent reaction to phosphatidylethanolamine (PE) embedded in the membrane (LC3-II) [36,37,38].

While the LC3 subfamily is predominantly involved in the elongation of the phagophore, gamma-aminobutyric acid (GABA)-receptor-associated proteins (GABARAPs) are important at the later stages of autophagosome maturation [41]. Both the LC3 subfamily (in humans, LC3A, LC3B, and LC3C) and GABARAP subfamily (GABARAP, GABARAPL1m and GABARAPL2/GATE-16) belong to the Atg8-family proteins. The Atg8-positive membranes will enable the formation of the double-layered autophagosomes [42] and engulf cargo targeted by autophagy adaptors. These adaptors possess LC3-interacting region (LIR) motifs, which bind the Atg8 proteins on the membranes. Terminally, the autophagosome content will be degraded upon fusion with lysosomes [43].

In mitophagy, specific mitophagy receptors mediate engulfment of damaged mitochondria by Atg8-positive membranes, as these receptors also contain LIR motifs [44]. Which Atg8 protein is lipidated might depend on the mitophagy pathway further upstream [45]. However, downstream lipidation results in recruitment of the Atg8-positive phagophores becoming mitophagosomes [46]. Where the lipids utilized for the mitophagosomal membranes arise from, is not clear. In 2013, autophagosomal membranes were found to be derived from MERCS [14]. However, a general loosening of MERCS is observed upon mitochondrial damage, raising controversy about the origin of mitophagosomal membranes [47].

Lastly, the mature mitophagosomes will fuse with lysosomes to degrade their cargo. Where the fusion occurs depends on the cell type as well as the cellular environment. Although mitochondria and lysosomes might be in contact at times, and lysosomes can promote fission, they are not reported to mediate mitochondrial degradation by this direct contact [48]. Lysosomes have a distinct distribution pattern in neurons, where many lysosomes are localized in the cell body, while fewer lysosomes are abundant in the long neuronal processes [49,50]. Therefore, specific mechanisms may be required to circumvent the spread of reactive oxygen species (ROS) that are produced by damaged mitochondria inside autophagosomes during their transport towards the cell body. Alternatively, the local fusion of mitophagosomes with axonal lysosomes has been observed [51], but the exact site of neuronal autophagosome–lysosome fusion is still a matter of discussion [52].

## 2. Mitophagy Pathways

Maintaining a pool of healthy mitochondria is essential for cellular survival. To maintain the quality of the mitochondrial network to prevent the spreading of ROS, defective mitochondria need to be isolated from the network and targeted for degradation. In contrast, healthy mitochondria fission, thereby they contribute to an extension of a healthy mitochondrial network. Dependent on the environment within the cell, mitophagy has to be balanced according to the current stimulus. Several mitophagy pathways have been described so far (Figure 1), which respond to different stimulations such as depolarization, nutrient deficiency, or hypoxia. In metazoans, mitophagy pathways may overlap to compensate for potential loss or defects in a parallel pathway, underscoring the importance of well-orchestrated mitophagy [19,53]. Furthermore, the prevalent mode of mitophagy might depend on the cell type, as there can be differences between post-mitotic and mitotic cells or in specialized cell types like erythrocytes that do not contain mitochondria [4].

### 2.1. The PINK1–Parkin Pathway Is a Ubiquitin-Dependent Mitophagy Pathway Induced by Mitochondrial Depolarization

The PTEN-induced kinase 1 (PINK1)–Parkin pathway is one of the best-characterized mitophagy pathways and is commonly activated upon mitochondrial depolarization (Figure 1A) [54]. Both PINK1 and Parkin are mutated in hereditary forms of Parkinson’s disease [55,56], fitting the general notion that mitochondrial quality control is an important determinant for neuronal health. Under homeostatic conditions, PINK1 is a protein with a short half-life of about 30 min [57]. Under physiological conditions, PINK1 is imported via the translocase of the outer membrane (TOM) and inner membrane (TIM) complex into the inner mitochondrial membrane, where it is cleaved by the rhomboid protease presenilin-associated rhomboid-like protein (PARL). Upon mitochondrial depolarization (which can, experimentally, be induced through the administration of compounds including CCCP, Antimycin A, and/or Oligomycin A), PINK1 degradation is reduced and it accumulates on the OMM [58]. Thereby, PINK1 initiates a signaling cascade, resulting in mitophagy [54].

Upon accumulation, PINK1 phosphorylates several proteins, including ubiquitin, Mitofusins (MFN), Miro, and the E3 ligase Parkin [59,60,61,62]. There, the phosphorylated proteins, most importantly, phospho-ubiquitin, facilitate Parkin translocation to the mitochondria [63,64,65,66,67,68]. Upon phosphorylation by PINK1, Parkin ubiquitinates OMM proteins, including voltage-dependent anion-selective channel 1 (VDAC1), MFN1, and MFN2 [47]. These proteins will be degraded via the proteasomal system, facilitating mitophagy [69]. The key autophagy-regulating ER protein, WIPI2, was recently found to be involved in the degradation of OMM proteins as well, next to its role in LC3 lipidation. Thereby, WIPI2 connects the early steps of mitophagy with the later stages of mitophagosomal uptake, but further implications and downstream mechanisms affecting mitophagy still need to be elucidated [70].

The poly-phospho-ubiquitin chains that accumulate on proteins of the OMM, upon Parkin activity, will recruit the autophagy adaptors optineurin (OPTN) and nuclear domain 10 protein 52 (NDP52, also known as CALCOCO2). These are activated, stabilized, and enhanced in their specificity by TANK-binding kinase 1 (TBK1) [71,72]. The activated autophagy adaptors will, in turn, recruit the Atg8-family proteins LC3 and GABARAP, which will lead to the engulfment of the damaged organelle by a phagophore [9,73]. Interestingly, it was found that the autophagy receptors OPTN and NDP52 can act independently of Parkin and mediate low-level mitophagy [73,74]. Recently, NDP52 was discovered to be a redox sensor, which could explain how this autophagy receptor responds to mitochondrial damage independent of recruitment through Parkin [75]. However, Parkin amplifies the mitophagy signal and leads to a more robust degradation of damaged mitochondria [73]. The autophagy receptor p62 (Sequestosome 1) was also found to promote Parkin-mediated mitophagy by binding ubiquitinated VDAC1 [5]. While it was reported that this autophagy receptor is required for Parkin-induced mitochondrial clustering [76], it is not essential for mitophagy, as its knockdown did not abolish mitophagy [77]. P62 rather plays a role in non-specific autophagic processes, supporting the idea that mitophagy pathways have multiple “safety nets” by which dysfunction in a specific pathway can be compensated by other, potentially less specific pathways.

Although, it is not known where the resulting mitophagosomal membranes derive from, as stated in the previous section, a recent study has shown that upon PINK1–Parkin-mediated mitophagy, mitochondria are involved in producing phosphatidic acid and diacylglycerol. These lipids are required for mitophagosome formation, suggesting a role for mitochondria-derived membranes [15].

### 2.2. Ubiquitin-Independent Mitophagy Pathways

Several proteins that mediate mitophagy were found to work independent of Parkin and are, thereby, also independent of the ubiquitination of OMM proteins. Many of these pathways overlap or can interfere with one another. The following sections will provide an overview about mitophagy-receptor-mediated pathways.

### 2.3. ULK1–FUNDC1 Mitophagy

ULK1 is involved in mediating mitophagy upon two different stimuli and, thereby, provides a great example on how the stimulus of induction influences the downstream mitophagy pathway. Upon nutrient deprivation, AMPK was found to activate ULK1 and its homolog ULK2, which then form a complex with FIP200, ATG13, and ATG101 [78,79]. This complex mediates the activation of the autophagy adaptor p62, which in turn recruits Atg8-positive membranes and leads directly to the degradation of mitochondria. This was the first established link between the nutrient status of the cell and mitophagy [29,80].

The second stimulus upon which ULK1 becomes activated by AMPK and forms a macromolecular complex, is hypoxia (Figure 1B) [81,82]. The ULK1–AMPK complex translocates to the mitochondria, where it interacts with Fun14 domain containing protein 1 (FUNDC1) [49,83]. FUNDC1 is an integral protein on the OMM and highly conserved in most mammals [5]. This mitophagy receptor is abundant on MERCS and can influence the contact between the two organelles [84]. It also possesses an LIR motif that directly binds LC3, independent of Parkin or ubiquitin [53,85]. How phosphorylation modulates FUNDC1 binding to LC3 has been discussed controversially [83,86], until the crystal structure of LC3 in complex with the LIR motif in FUNDC1 identified the phosphorylation of Tyr 18 within the LIR as a negative regulator for LC3 binding [87]. Besides binding LC3, FUNDC1 promotes mitophagy by altering mitochondrial dynamics. It interacts with optic atrophy 1 (Opa1) and dynamin-related protein 1 (Drp1), causing increased fission [88].

Increased fission is an important common motif preceding mitophagy, as seen by the redundancy across the different pathways. Although FUNDC1 mitophagy is independent of the E3 ligase Parkin, finetuning is mediated by another E3 ligase, the membrane-associated RING-CH (MARCH5). MARCH5 ubiquitinates FUNDC1, which reduces mitophagy [89]. On the other hand, MARCH5 can also promote mitophagy by controlling Drp1 and, thereby, influencing mitochondrial fission [90,91]. How MARCH5 activity is controlled in mitophagy-inducing conditions is not clear yet and remains to be elucidated.

### 2.4. AMBRA1 Mitophagy Receptor

Under basal conditions, AMBRA1 is present at the mitochondria and MAM domains of the ER, where it was described to induce non-specific autophagy [92,93], in addition to its role in the initiation of general autophagy [94]. On mitochondria, the pro-autophagic activity of AMBRA1 is inhibited by (B-cell lymphoma 2) BCL-2 [95]. However, upon depolarization of the mitochondria, AMBRA1 interacts with Parkin as well as stabilizes PINK1 by interacting with ATPase Family AAA Domain Containing 3A (ATAD3A) [96,97]. Thereby, AMBRA1 mediates mitophagy in a Parkin-dependent manner in vitro and in vivo [97,98]. Additionally, AMBRA1 possesses an LIR motif and is also involved in Parkin-independent mitophagy [99].

Upon oxidative stress caused e.g., by ischemia, AMBRA1 is posttranslationally activated by the phosphorylation of Ser1014, via the IκB Kinase α (IKKα), which, in turn, is controlled by the E3 ligase HECT, UBA, and WWE Domain Containing E3 Ubiquitin Protein Ligase 1 (HUWE1). Phosphorylation alters the structure of AMBRA1, resulting in increased interaction with Atg8-family proteins and subsequent mitophagic activity (Figure 1C). This stimulation of mitophagy occurs independently of the main mitophagy receptors and p62 [99,100]. Besides phosphorylating AMBRA1, HUWE1 is a crucial partner for AMBRA1 by mediating the ubiquitination of MFN2 on the OMM. As already described in the PINK1–Parkin-mediated mitophagy pathway, MFN2 is becoming degraded and, thereby smoothens the way for the required fragmentation of the mitochondria preceding mitophagy [100].

While most of the finetuning mechanisms behind AMBRA1-mediated mitophagy are not described yet, induced myeloid leukemia cell differentiation protein 1 (MCL-1), a member of the BCL-2 family, was recently found to delay AMBRA1-dependent mitophagy [101]. Fittingly, MARCH5 was reported to act upon MCL-1, establishing an interesting link between FUNDC1- and AMBRA1-mediated mitophagy. Upon the loss of MARCH5, the levels of the antiapoptotic protein MCL-1 were increased [102], providing an intriguing base for further experiments exploring the interconnection between the different mitophagy regulators.

### 2.5. BNIP3/NIX Mitophagy Receptor

BCL-2 interacting protein 3 (BNIP3) and Nip-like protein X (NIX, also known as BCL-2-interacting protein 3 like, BNIP3L) are proapoptotic BH3 proteins and members of the BCL-2 family. They are located on the OMM and, as with the other autophagy receptors, these proteins contain LIR motifs [103]. Upon hypoxia, BNIP3 and NIX levels increase, mediated by the transcription factor hypoxia-inducible factor-1 (HIF1α), and mitophagy is induced [104,105]. Upon moderate hypoxia conditions (10% O_2_), BNIP3 was found to be phosphorylated by c-Jun N-terminal kinase (JNK) 1/2 at Ser60 and Thr66, what stabilizes and activates the protein, resulting in the recruitment of LC3 and mitophagy initiation (Figure 1C). In contrast, under severe hypoxia conditions (0.3% O_2_), JNK1/2 becomes inactivated and protein phosphatases (PP) 1/2A dephosphorylate BNIP3 at Ser60/Thr66, which accelerates BNIP3 proteasomal degradation [105]. It is tempting to speculate that the phosphorylation of BNIP3 in Ser60/Thr66 is a molecular mechanism of mitochondria to sense hypoxia within the cell. Potentially, this mechanism can be considered a molecular switch between protective mitophagy upon mild damage and protective autophagy or even cell death in response to severe damage through hypoxia, as BNIP3 was shown to be involved in all of these processes [106,107]. However, a link between these phosphorylation sites and the downstream protein function still waits to be explored.

Another protein able to stabilize and activate BNIP3 upon phosphorylating Ser17 is ULK1. Again, LC3-BNIP3 binding is promoted, resulting in mitophagy initiation [108]. The phosphorylation of Ser17 and Ser24 flank the LIR motif of BNIP and have already been shown to mediate mitophagy or induce apoptosis [109], but whether they also behave in a switch-like manner is yet unknown.

ULK1 also phosphorylates the mitophagy receptor NIX. NIX was initially recognized for its essential function in the removal of mitochondria in erythrocytes and reticulocytes [110,111,112]. These are also the first established links between NIX and mitophagy. However, by now, NIX, the proautophagic protein that contains two LIR motifs, is recognized to be involved in far more processes. The LIR motif near the N-terminal is conserved in both BNIP3 and NIX. The phosphorylation of this LIR motif enhances the binding affinity to LC3 and, thereby, promotes mitophagosome recruitment to mitochondria (Figure 1C). Upon the lack of phosphorylation, the NIX-LC3 binding affinity is low, compared to the other Atg8 proteins [113,114,115]. The second LIR motif is localized near the C-terminal and the BH3-like domain [113]. With this C-terminal LIR motif, NIX can recruit GABARAPL1 to damaged mitochondria. However, details of downstream pathways such as the recruitment of Atg proteins or other autophagy adaptors are not yet reported.

Few studies have reported links between PINK1–Parkin mitophagy and NIX. However, the evidence is mixed on whether NIX acts in parallel or within the PINK1–Parkin pathway. In 2010, it was reported that NIX, along with Parkin, was involved in mitochondrial priming and the controlled translocation of Parkin to the mitochondria [116], suggesting that NIX acts within the PINK1–Parkin pathway. However, most studies report a compensatory role, in which upregulated NIX levels might compensate to a certain level for the loss of Parkin in human cells [117,118]. Interestingly, in *Drosophila melanogaster*, it was shown that only the knockdown of *pink1* but not the loss of *park* (Parkin) could be rescued by NIX [119]. Lastly, NIX was found to be a substrate of Parkin. The ubiquitination of NIX recruits the neighbor of BRCA1 (breast cancer type 1) (NBR1) to mitochondria and, thereby, targets the organelle for degradation [119].

In summary, there is extensive crosstalk between the different mitophagy pathways, with some species- or cell-type specific differences.

## 3. Tether Proteins and Their Characteristics

Throughout the past decades, there has been a steady increase in reports on organellar contact sites, with mitochondria being found to contact every other membrane-bound structure in the cell [120,121]. About 20% of the mitochondrial surface is thought to be associated with ER membranes [122]. While the regions of the ER membranes associated with mitochondria are often referred to as mitochondria-associated membranes (MAMs), the term MERCS refers to the entire proteome abundant on interconnected mitochondria and ER. MERCS can be established between mitochondria and rough as well as smooth ER. The distance between rough ER and mitochondria was previously observed to be around 30–65 nm [1], while smooth ER MERCS are described to be tighter. These contacts are between 10–30 nm during the resting condition [123] and might tighten to 10 nm upon environmental stress [124]. Although often not commented upon, most studies presumably refer to the MERCS with smooth ER, as lipid and ion transfer mostly occurs between smooth ER and mitochondria, while rough ER MERCS are not as well-described [125]. Due to lack of clarification in the original literature, this review will not discriminate between rough and smooth ER MERCS.

As with all inter-organellar contact sites, MERCS exhibit a specific proteome and lipidome that are required for the interplay between the organelles. Some of the proteins required to maintain inter-organellar interactions are structural proteins such as tethering proteins, which mediate the organellar interaction (Figure 2A), as well as spacer proteins, that define the distance between the two organelles. [125]. To be characterized as a mitochondria–ER tether, the proteins need to fulfil three criteria [120]. First, one of the proteins needs to be abundant on the OMM, while its homo- or heterotypic interaction partner is abundant on the MAMs of the ER. Second, manipulation of any of the proteins will lead to altered MERCS abundance or spacing, e.g., by increasing the distance between mitochondria and ER upon deletion. Lastly, functional implications must follow, which can include the reduction of Ca^2+^ or phospholipid trafficking between mitochondria and ER [120,121,126]. Some functions are regulated by several tether pairs, like Ca^2+^ signaling [127,128,129], while other MERCS-associated functions, including mitochondrial trafficking, are mainly regulated by a singular tether pair [130]. Other processes, like the de novo synthesis of sphingolipids, cannot be assigned to specific tether proteins, but rather to other proteins abundant at the MERCS [131]. The following sections will describe the thus far best-characterized MERCS in metazoans and their specific function, if known.

### 3.1. IP3R–VDAC1

Initially, MERCS were thought to have one major function: the transfer of Ca^2+^ from the ER storage to the mitochondria. The ER channel protein inositol 1,4,5-trisphosphate receptor (IP3R) and VDAC1 located on the OMM form a functional connection that is mostly known for Ca^2+^ transfer from the ER to mitochondria. Notably, both IP3R and VDAC1 are abundant in different isoforms within different cell types [132,133,134,135]. However, studies often do not discriminate between the different IP3R isoforms (IP3R1, IP3R2, and IP3R3), whereas VDAC1 is the isoform that is most studied in the context of MERCS as it is the only isoform present in co-immunoprecipitates with IP3R receptors [136]. Therefore, this review will focus mainly on VDAC1.

IP3R and VDAC1 are functionally linked by Mortalin (also known as the chaperone glucose-regulated protein 75 (GRP75), or mitochondrial heat shock protein 70 (mtHSP70)), serving as an adaptor protein between the two transmembrane proteins (Figure 2A) [128]. As a chaperone, Mortalin serves a variety of functions in several subcellular locations, from the mitochondrial matrix to the nucleus [137], complicating studies that aim to decipher its role in MERCS. Also, a fourth binding partner has been suggested: the Parkinson’s-disease-associated protein DJ-1 was shown to interact with the IP3R–VDAC1 tether to regulate its function [138]. The deletion of the ER channel IP3R leads to reduced mitochondrial Ca^2+^ uptake and a loss of bioenergetic coupling, provided by the stimulation of enzymes of the citric acid cycle [139]. However, the recruitment of IP3R at MERCS upon mitochondrial proximity by an inducible synthetic linker still enhanced mitochondrial Ca^2+^ uptake [140]. Therefore, an intramolecular interaction between IP3R and VDAC1 for efficient Ca^2+^ transfer may not be necessary, and other MERCS could increase mitochondrial Ca^2+^ uptake as a secondary effect.

### 3.2. MFN2–MFN2/MFN2–MFN1

MFN2 is mostly known for its function in mitochondrial fusion, mediating co-operation across the gap of two adjacent mitochondria to allow their joining into a single organelle [141]. Studies by Luca Scorrano and colleagues have revealed that MFN2, but not MFN1, also localizes to the ER membrane [142], thereby establishing MERCS through interaction with its mitochondrial isoform. In contrast, MERCS mediated by homodimerized MFN2 do not induce fusion events, yet are important for proper ER morphology and the transfer of Ca^2+^ ions from the ER to mitochondria [142]. MERCS are reduced in MFN2-deficient cells [143], and, the distance between mitochondria and ER is increased [126]. Interestingly, mutations in MFN2 but not MFN1 are causative of the peripheral neuropathy Charcot–Marie–Tooth type 2A [144], in line with an additional function of MFN2 for neuronal homeostasis.

### 3.3. VAPB–RMND3

Vesicle-associated membrane protein B (VAPB) is located to the ER and interacts with Regulator of Microtubule Dynamics 3 (RMDN3), which is also known as Protein tyrosine phosphatase-interacting protein-51 (PTPIP51) and resides on the OMM. The proteins interact via an FFAT motif (RMDN3) and an MSP domain (VAPB). However, a recent study has shown that the coiled coil domain nearby the FFAT motif of RMDN3 is essential for binding to VAPB [145]. The modulation of the VAPB–RMND3 contact site can influence Ca^2+^ transfer from the ER to mitochondria by acting on IP3R [145]. Interestingly, for this particular tether, a role in phospholipid transfer has been described (Figure 2A) [146,147,148]. The ER-localized oxysterol-binding protein-related proteins 5 (ORP5) and ORP8 have been shown to interact with RMDN3 at MERCS [149]. ORP5/8 mediate the exchange of PI(4)P and phosphatidylserine (PS) from the plasma membrane to the ER [150], coupling the VAPB–RMDN3 contact site to the metabolism and the local availability of phosphatidylinositol phosphates (PIPs), which are also involved in the activation of the general autophagic machinery. Furthermore, several interactions with components of the autophagic machinery have been reported and will be discussed below, suggesting an intriguing crosstalk between lipid biogenesis and the VAPB–RMND3-mediated MERCS involved in phospholipid transfer.

Mutations in VAPB cause familiar forms of amyotrophic lateral sclerosis [151,152]. Additionally, the amyotrophic-lateral-sclerosis-associated, aggregation-prone RNA-binding protein TAR DNA-binding protein 43 (TDP-43) was reported to alter VAPB–RMND3 interaction [153], suggesting a convergence of several pathways for neurodegeneration in amyotrophic lateral sclerosis involving this tether pair. Fitting to their role in neuronal health, the interaction between VAPB and RMDN3 also affects neurons by regulating synaptic function and dendritic spine morphology [154]. How these neuron-specific functions tie to the role of VAPB and RMDN3 in mediating phospholipid and Ca^2+^ transfer from the ER to mitochondria, and whether the interaction of the tether proteins prevent or induce neurodegenerative pathways, will be interesting to discover in the future.

### 3.4. BAP31–FIS1

The B-cell receptor-associated protein 31 (BAP31) and mitochondrial fission protein 1 (FIS1) tether pair connecting mitochondria and ER were first described in 2011 and were found to be involved in apoptotic signaling by activating procaspase 8 (Figure 2A) [155]. Due to the known role of FIS1 as a DRP1-adaptor protein [152], the overexpression and knockdown of FIS1 alters mitochondrial dynamics by leading to elongated or fragmented organelles. However, the depletion of FIS1 does not alter MERCS [156]; hence, FIS1 and BAP31 cannot be defined as a true tether if a strict definition of tether pairs is applied. However, it is intriguing to consider that contacts to the ER define future sites of mitochondrial fission [157], which may partially be dependent on this tethering pair. Thus, this tether pair might be involved in mediating mitophagy by influencing mitochondrial dynamics, a common motif preceding mitophagy.

Deletion of the multifunctional sorting protein phosphofurin acidic cluster sorting protein 2 (PACS-2) causes BAP31-dependent mitochondrial fragmentation, also caused by controlling the apposition to the ER. This specific type of mitochondrial fragmentation seems to precede apoptotic signaling via the activation of the proapoptotic BCL-2 protein Bid at MERCS [158].

While BAP31 has not yet been found to be involved in human diseases, FIS1 interaction with DRP1 was increased in Alzheimer’s disease patients. Thereby, mitochondrial fission presumably contributes to the pathogenesis of Alzheimer’s disease [159]. How much the secondary role of FIS1 as a tethering protein is relevant to these neuropathological findings, remains to be determined.

### 3.5. RRBP1–SYNJ2BP

In 2017, Ribosome binding protein 1 (RRBP1 also known as p180) and Synaptojanin 2 binding protein (SYNJ2BP) were identified as novel tether mediating MERCS, using a proximity biotinylation approach [160]. RRBP1 is localized to the ER and was stated to be involved in the UPR in response to ER stress [161]. The accumulation of misfolded proteins in the ER, often induced by treatment with drugs such as tunicamycin [162,163], leads to the activation of several pathways that attenuate general protein translation and induce the expression of chaperones, including ER chaperones [164,165]. As the name suggests, RRBP1 is associated with ribosomes and is also known to bind mRNA to locate RNA to the ER in a translation-independent manner [159]. Not surprisingly, RRBP1–SYNJ2BP were suggested to be involved in protein translation and is the only known tether to increase contacts specifically with the rough ER (Figure 2A) [160]. In liver cells, these tethering proteins cause the formation of mitochondria tightly wrapped in ER, termed WrappER. Thereby, RRBP1–SYNJ2BP regulate the generation of lipoproteins [1].

In neurons, it is unclear how much of this contact site is present throughout the cell, specifically in axons that are generally thought to be devoid of rough ER. However, in pathophysiologic conditions, the increased interaction between SYNJ2BP and RRBP1 was shown to disrupt mitochondrial distribution within the cell [166], pointing towards a role for this tether pair also in neurons.

### 3.6. PDZD8

PDZ domain containing protein 8 (PDZD8) is an ER-resident protein that was recently identified as part of a mitochondria–ER tether (Figure 2A). It is abundant at MAMs and contains a synaptotagmin-like mitochondrial-lipid binding protein (SMP) domain, predisposing it to mediate lipid transfer at MERCS [167]. Fittingly, PDZD8 was found to be involved in phospholipid transfer between the two organelles [168]. Upon the knockdown of PDZD8 in *D. melanogaster* and mammalian cell lines, MERCS were decreased in number, fitting to the characteristics of a true tether protein [167,169]. In mammalian neurons and NIH3T3 cells, the knockdown of PDZD8 was found to reduce Ca^2+^ transfer from ER to mitochondria, although the release of Ca^2+^ out of the ER was not affected [167]. Since expression of an artificial mitochondria–ER tether rescued this phenotype [167], it is likely that PDZD8 is not directly required for Ca^2+^ transfer, but rather as a secondary mechanism to maintain structure of MERCS. Lastly, PDZD8 was found to interact with late endosomes via Rab7, and this contact recruits mitochondria as well, forming a tripartite contact between the three organelles [170]. However, it still remains to be elucidated what the mitochondrial interaction partner of PDZD8 is and how its function regulates MERCS in health and disease.

## 4. Crosstalk between Mitophagy and MERCS

It is unclear, whether the initiation of mitophagy requires the loosening of MERCS to provide space for the growing phagophore engulfing the segregated and damaged mitochondria, or if some contact sites need to be maintained during mitophagy as a phospholipid source for phagophore growth [47,171,172]. In the following chapters, we will summarize the current knowledge on the response of MERCS to mitochondrial damage and their potential involvement in the regulation of mitophagy.

### 4.1. MFN2, VDAC1, and SYNJ2BP Are Diminished upon PINK1–Parkin Activation

As described above, Parkin localizes to damaged mitochondria where it ubiquitinates multiple OMM proteins, serving as receptors for autophagy adaptor proteins. Among these ubiquitinated OMM proteins are several mitochondrial tether proteins mediating MERCS, including MFN2, VDAC1, and SYNJ2BP (Figure 2B) [63,173,174]. MFN2 is phosphorylated by active, OMM-localized PINK1, and thereby primed for ubiquitination by Parkin [63]. Ubiquitination then leads to the removal of MFN2 from the OMM through rapid turnover by the proteasome (Figure 2B) [69]. The loss of MFN2 has been suggested not only to enable the isolation of damaged mitochondria by restricting mitochondrial fusion, the main function of MFN2, but also to uncouple the damaged organelle from the ER [47]. A similar case may occur in Parkin-independent mitophagy, mediated by AMBRA1- and HUWE1-dependent mitophagy [100]. Indeed, the induction of mitochondrial damage with the uncoupler CCCP in U2OS cells reduced the observed contacts between mitochondria and ER in electron microscopy, in both, a Parkin-dependent and independent fashion [47]. However, this analysis included mostly intact mitochondria that were not yet being engulfed by autophagosomes. Therefore, it is possible that the interaction between mitochondria and ER is altered at later stages, which would be in line with the observation that the proteasomal degradation of OMM proteins precede the formation of autophagosomes [175], but that still remains to be confirmed.

The ubiquitination of VDAC1 and SYNJ2BP by Parkin leads to their degradation via the proteasome and, presumably, results in a loosening of the MERCS (Figure 2B). For VDAC1, an interesting distinction dependent on the length of the ubiquitin chain on the protein channel was reported. While poly-ubiquitination promotes mitophagy through the PINK1–Parkin pathway, monoubiquitinated VDAC1 induces apoptosis [176]. It is interesting to speculate that poly-ubiquitination favors the proteasomal destruction of the VDAC1-mediated MERCS, whereas mono-ubiquitination may be insufficient to promote the efficient extraction of this beta-barrel OMM protein from the membrane. Instead, mono-ubiquitination could stabilize the VDAC1-mediated MERCS and enhance mitochondrial Ca^2+^ uptake, causing mitochondrial Ca^2+^ overload and cell death. How VDAC1 mono-ubiquitination would influence its binding to Mortalin or IP3R remains to be determined. Fittingly, the IP3R-mediated Ca^2+^ transfer from the ER to mitochondria is not needed for efficient Parkin recruitment to damaged mitochondria. However, IP3R may be a prerequisite for the mitochondrial clustering observed downstream of Parkin recruitment [177], yet the role of this clustering for mitophagy is unknown.

Recently, a novel role for SYNJ2BP in PINK1–Parkin-mediated mitophagy has been described in neurons. Here, the short half-life of the PINK1 protein limits its availability to mitochondria located distal from the cell body. To enable PINK1 synthesis and stabilization at distal mitochondria, the *Pink1* mRNA was found tethered to the mitochondrial outer surface and is thereby transported into the distal parts of the neuron [178]. Interestingly, this co-transport of *Pink1* with mitochondria depends on SYNJ2BP in concert with a specific isoform of the lipid-phosphatase SYNJ2, a different PDZ-motif-containing protein [179], which also contains an RNA-binding domain [178]. Both, RRBP1 and SYNJ2 presumably bind the PDZ domain of SYNJ2BP in the same manner, suggesting that the formation of MERCS via SYNJ2BP-RRBP1 might interfere with the efficient transport of *Pink1* mRNA. However, the connection between SYNJ2BP and mRNA transport still needs to be experimentally proven. Previous studies have shown that, like MFN2 and VDAC1, SYNJ2BP is a Parkin substrate [174,175], suggesting that not only will the MERCS mediated by SYNJ2BP and RRBP1 be released upon Parkin activation, but also the tethering of the *Pink1* mRNA could be diminished. Evidence from our lab suggests that release of the *Pink1* mRNA from SYNJ2BP and mitochondria stimulates PINK1 production and, thereby, promotes mitophagy [180].

In conclusion, the abolishment of RRBP1–SYNJ2BP-mediated MERCS through the degradation of SYNJ2BP could serve two functions: First, by loosening the binding between the mitochondria and ER, SYNJ2BP allows mitochondrial engulfment; and second, local production of PINK1 is stimulated through the release of RNA from SYNJ2BP, thereby recruiting Parkin in a positive feedback loop.

### 4.2. RRBP1 Binding Stimulates LC3 Lipidation at MERCS

RRBP1 was recently associated with the regulation of mitophagy in a pathway parallel to PINK1–Parkin-dependent mitophagy. Upon depolarization, resulting in protein import stress, the mitochondrial matrix protein Nod-like receptor protein NLRX1 fails to be imported into mitochondria via the TOM complex and remains in the cytosol, where it associates with RRBP1 (Figure 2C) [181]. NLRX1 contains a putative LIR motif [182], which then recruits LC3 to RRBP1 and hence to MERCS. Thereby, the NLRX1–RRBP1 complex controls LC3 lipidation at the site of mitophagosome formation. Although this mechanism acts independently of PINK1–Parkin activation, it potentiates the efficiency of mitochondrial removal by PINK1–Parkin-dependent mechanisms. Intriguingly, the same study also suggested that RRBP1 acts as a sensor for both, translational stress, potentially due to its ability to interact with ribosomes [159], as well as for mitochondrial import stress by a yet-to-be-determined mechanism [181]. Both, translational and protein import stress lead to the formation of an SDS-insoluble, high-molecular-weight (HMW) form of RRBP1. This HMW form of RRBP1 interacts with the RNA-binding protein splicing factor, proline- and glutamine-rich (SFPQ), which in neurons delivers transcripts encoding various mitochondrial proteins into the axon [183]. Together, RRBP1, SFPQ, and NLRX1 form a ternary complex [181], in which, presumably, SYNJ2BP, the RNA tether and mitochondrial partner of RRBP1, also takes part [184].

The binding of RNAs to SYNJ2BP protects nuclear-encoded mitochondrial mRNAs during translational stress to enable a quicker recovery of mitochondrial function upon the release of translational repression [184]. A similar effect was reported for RRBP1 [181], suggesting that this protection may occur at MERCS. Therefore, it is tempting to speculate that the local translation of some mRNAs at the mitochondrial surface is concentrated at RRBP1–SYNJ2BP-mediated MERCS, which may shield the mRNAs from degradation upon stress or even global changes in translational regulation. A recent preprint suggests that translational repression due to ER stress sensing is modulated at MERCS formed by a putative tether pair—ATPase Family AAA Domain Containing 3A (ATAD3A) at the mitochondria, and the ER stress-sensing kinase PERK on the other organelle. This contact allows the continued translation of mitochondrially localized ribosomes, despite global translational repression due to activation of the UPR [185]. It remains to be determined if these interacting proteins are a true, functional tether or if the interaction may be induced only downstream of RRBP1–SYNJ2BP tethering. Additionally, more work is required to assess how RRBP1–NLRX1 complex formation affects the regulation of PINK1 biogenesis in neurons, considering the above-mentioned role of SYNJ2BP in *Pink1* mRNA transport. Most likely, future research will reveal a multi-layered response system equipped to cope with mitochondrial protein import stress, from the induction of the UPR to the initiation of mitophagy as a last resort to eliminate the damaged organelle.

### 4.3. VAPB–RMND3 Contacts Regulate Autophagy

As alluded to before, VAPB has been reported to directly interact with ULK1 and FIP200, core components of the general autophagic machinery, mediating autophagosome formation [186]. Furthermore, the overexpression of either VAPB or RMDN3 reduces autophagy, whereas their knockdown has the opposite effect [187]. Specifically, autophagy driven by the inhibition of mTOR signaling was affected, but not starvation-induced autophagy. Mechanistically, the authors suggested that this effect underlies increased Ca^2+^ signaling in the mitochondria [187]. As the reduction in autophagy can be rescued by the expression of a synthetic tether, altered autophagy levels might not be specific to the action of VAPB–RMND3 but could involve any tether that favors IP3R–VDAC1 apposition and thereby increases mitochondrial Ca^2+^ uptake. Hence, it is tempting to speculate that this may be driven by mitochondrial ATP output, stimulated by Ca^2+^ influx [188]. Increased ATP output may decrease the activation of AMPK, which, in turn, will activate mTOR and prevent general autophagy pathways [189]. How VAPB and RMDN3 are involved in more specific forms of mitophagy, remains to be determined.

As RMDN3–VAPB play a unique role in mediating phospholipid exchange across the MERCS, it is tempting to hypothesize that upon induction of mitophagy, this contact site need to be maintained to allow the flow of phospholipids to fuel the formation of the phagophore. A similar role could be played by PDZD8-mediated MERCS, due to its role in lipid transfer [168]. Fittingly, the knockdown of PDZD8 decreases mitophagy [167,169]. However, it remains to be elucidated if either or both of these contact sites provide some or all of the phagosomal lipids.

### 4.4. FIS1 Mediates Mitophagy via Regulating STX17 Localization

The role of FIS1 in mitophagy seems to be multifaceted. Not only are asymmetric fission events coupled to FIS1-dependent DRP1 recruitment [24], but also other effectors of degradative pathways rely on this protein for their coupling to mitochondria. The SNARE protein STX17 is involved in autophagosome formation, as well as mediating lysosome–autophagosome fusion [14,190]. Furthermore, STX17 can shuffle between mitochondria and ER. Whether this shuffling requires FIS1–BAP31 contact sites remains to be determined. However, it does seem highly likely, as, upon starvation-induced autophagy, STX17 was discovered to relocate to MERCS [14]. Upon FIS1 knockdown, STX17 remains trapped on the OMM and its self-oligomerization triggers mitophagy (Figure 2D) [191].

Recently, STX17 was reported to possess two LIR motifs which enable it to mediate mitophagy independent of PINK1 and Parkin [192]. STX17 can also bind ATG14, a protein required for the early steps of autophagosome formation, and, thereby, STX17 recruits this PI3K to MERCS during starvation [14]. This induces the local generation of PI3P, a signal necessary for the formation of autophagosomes [193]. The scaffolding function of STX17 is independent of its SNARE domain [194], unlike its later role in the targeting of the autophagosome to the lysosome [190]. Additionally, STX17 regulates the OMM localization of the FUNDC1 phosphatase phosphoglycerate Mutase Family Member 5 (PGAM5), thereby facilitating FUNDC1-dependent mitophagy [195].

Analogous to its role in autophagosome–lysosome fusion, STX17 mediates the delivery and the subsequent fusion of mitochondria-derived vesicles (MDVs) to the late endosomes/lysosomes through its SNARE domain. Damaged mitochondrial material is thereby degraded in a pathway separate from canonical mitophagy, which would include the hierarchical processing of the mitophagosomes [196]. This function of STX17 requires the activation of the PINK1–Parkin pathway, but is independent of the general autophagic machinery [197]. It remains to be determined if MERCS also regulate MDV generation, and how STX17-mediated MDV generation is regulated by changes in BAP31–FIS1 abundance. It is intriguing to speculate that the presence or absence of MERCS may determine the degradative pathway that STX17 initiates.

## 5. Conclusions and Outlook

To date, not all proteins mediating contact between mitochondria and ER are known, and future research will most certainly reveal other tethering proteins, such as the unknown binding partner of PDZD8 on the OMM or the potential contact site between ATAD3A and PERK mentioned above. Fitting to a role for MERCS in mitophagic regulation, ATAD3A was found to be associated with the regulation of mitophagy as it interacts and upregulates PINK1 [198]. Also, ATAD3A was suggested to bind binding immunoglobulin protein (BIP, also known as GRP78) and thereby suppress ER stress [199,200]. Yet, this putative tether pair requires validation as to whether changes in its abundance will alter the number or extent of MERCS.

The induction of mitophagy is emerging as yet another function of MERCS, next to their traditional roles in Ca^2+^ and phospholipid transport. Specifically, the role of MERCS as initiation sites for the autophagosome is an intriguing concept [14,201]. However, how this can be reconciled with the general loosening of MERCS upon mitophagy-inducing conditions [47] still needs to be elaborated. Presumably, some tethers, for example, MFN2–MFN2, IP3R–VDAC1, and possibly also RRBP1–SYNJ2BP, are targeted by the degradative cascade upon PINK1–Parkin mitophagy activation, to provide space for the membrane formation and organellar engulfment. In contrast, other tether interactions, for example BAP31–FIS1, VAPB–RMND3, and PDZD8, with its respective tether on the mitochondria, could persist and serve as initiation sites for the mitophagosome. Alternatively, selective degradation of tether proteins on the OMM might depend on the stimulus inducing mitophagy as well as its following mechanism. As PINK1, in association with Beclin 1, is localized to MERCS, where PINK1 promotes the formation of the phagophore [172], MERCS seem to play a role in PINK1–Parkin-mediated mitophagy. Furthermore, while STX17 is not necessary for the initial Parkin translocation to the mitochondria, it is required for the expansion of Parkin-mediated ubiquitin labelling across the entire mitochondria—a process that might also require MERCS [195].

The contacts between mitochondria and ER are the locations for the synthesis of sphingolipids, and contain high levels of these lipids, as well as they are rich in cholesterol [131,202,203]. Recently, cholesterol levels have been tied to PINK1 biology and to defective mitophagy in Alzheimer’s disease [204,205]. Future research will determine if the lipid composition at MERCS is important for the formation of the mitophagophore and how this interplays with the proteins mediating mitophagy initiation.

## Figures and Tables

**Figure 1 biomolecules-13-01198-f001:**
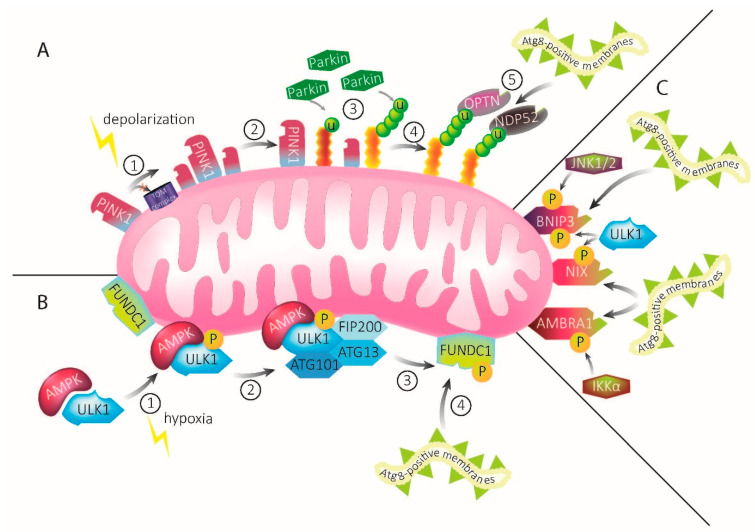
Mechanisms of mitophagy. (**A**) PINK1–Parkin-mediated mitophagy. Under homeostatic conditions, PINK1 is constantly imported into mitochondria. Upon mitochondrial damage, import of PINK1 via the TOM complex and subsequent degradation is inhibited and the protein accumulates on the OMM (1). Accumulated PINK1 recruits Parkin (2), which ubiquitinates OMM proteins (3). Ubiquitination recruits the autophagy receptors OPTN and NDP52 (4). The autophagy receptors, in turn, recruit Atg8-positive membranes, leading to engulfment of the damaged mitochondria into mitophagosomal membranes (5). (**B**) ULK1–FUNDC1-mediated mitophagy upon hypoxia. Upon mitochondrial damage through hypoxia, AMPK activates ULK1 via phosphorylation (1), leading to the recruitment of FIP200, ATG13, and Atg101 (2). Together, they form the ULK1–AMPK complex. This complex interacts with the mitophagy receptor FUNDC1 that is abundant on mitochondrial membranes, leading to its phosphorylation (3). FUNDC1 possesses an LIR motif that recruits Atg8-positive membranes (4). These will eventually engulf the damaged mitochondria. (**C**) The mitophagy receptors BNIP3, NIX, and AMBRA1 possess LIR motifs which directly bind and recruit Atg8-positive membranes upon activation by signaling pathways, including ULK1, IKKα, and JNK1/2.

**Figure 2 biomolecules-13-01198-f002:**
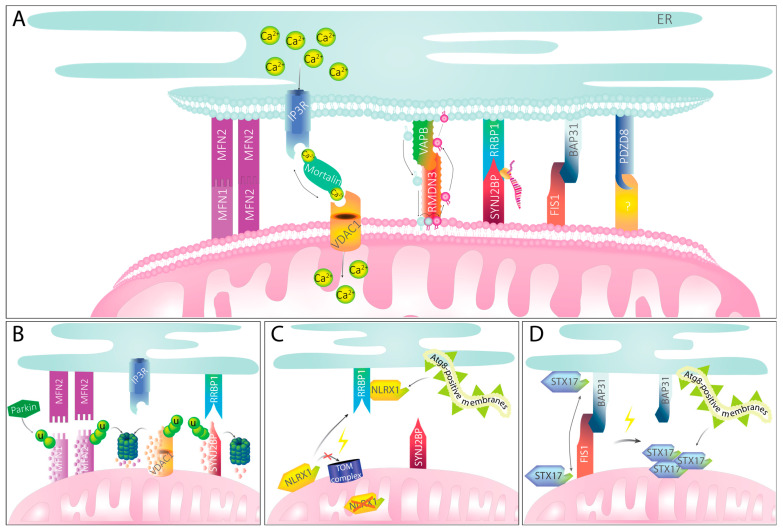
MERCS and their implications in mitophagy. (**A**) Mitochondria–ER tether proteins maintain physical contact between the organelles and mediate MERCS-specific functions. For selected tether pairs, their function in Ca^2+^ or phospholipid transport or their role in localized translation is exemplarily shown. (**B**) Mitochondria–ER tether proteins upon PINK1–Parkin-mediated mitophagy induction. Recruitment of Parkin through PINK1 accumulation leads to ubiquitination of mitochondrial partners of MERCS tethers. MFN2, VDAC1, and SYNJ2BP become ubiquitinated and subsequently degraded by the proteasome. Tethering to the ER is presumably getting ablated, providing space for engulfment of mitochondria. (**C**) Mitophagy mediated by NLRX1 signaling. Upon mitochondrial damage, NLRX1 cannot be imported into the mitochondria via the TOM complex and, instead, remains stable in the cytosol. There, it binds RRBP1. With its LIR motif, NLRX1 recruits Atg8-positive membranes that will eventually engulf damaged mitochondria. (**D**) Mitophagy mediated by FIS1 and STX17. Under homeostatic conditions, STX17 can shuffle between mitochondria and ER. Upon ablation of FIS1, STX17 remains on the mitochondria and self-oligomerizes. With its two LIR motifs, STX17 can recruit Atg8-positive membranes, mediating mitophagy.

## Data Availability

Not applicable.

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
