# Peer review of "Role of Mitochondria–ER Contact Sites in Mitophagy"

_biomolecules, 2023, doi:10.3390/biom13081198_

Round 1
Reviewer 1 Report
In this review Ruhmkorf and Harbauer discuss the role of the mitochondria-ER contact sites (MERCS)in the mitophagy. They started discussing the mechanisms that preserves the mitochondrial health and how these are linked to autophagy. then they discuss the main pathways of mitochondrial autophagy (mitophagy). They continue by describing the importance of MERCS that is the molecular participants of this event and the processes in which they are involved to finally conclude their work.
The work is well written and easy readable. the figures content and quality is adequate and the work has elevated significance to the field...
In my opinion the paper is suitable for publication upon very few changes.
as minor comments:
L15 the authors introduce to an acronym for mitochondria-ER contact sites (MERCS). I am personally more familiar with a similar term coined earlier (MAMs) referring to Mitochondrial associated membranes. As a recommendation, the authors may want to include a brief sentence indicating if such terms can be used for the sites of interactions of mitochondria with ER or carefully indicate their differences.
L341: the calcium transfer is always from ER to mitochondria and not between them indistinctly. It must be clearly stated.
L350 “However, the release of Ca2+ ions via IP3R upon mitochondrial proximity and mitochondrial stimulation is independent of the presence of a corresponding VDAC1 molecule in the OMM” this is not clear. As the authors know, VDAC1 does not stimulate the release of calcium by the IP3R. The IP3R is a ligand activated channel. If the authors wanted to mean that ER-mitochondrial transfer via IP3 release is not dependent of VDAC1, it is truth, and they may wan to discuss the presence of other VDAC isoforms.
finally, all the genes and Latin terms (in vivo, in vitro, e.g., et al., ) must be in italics.
Author Response
We would like to thank this reviewer for the kind words and helpful suggestions. Here is the point by point response to the minor comments:
L15 the authors introduce to an acronym for mitochondria-ER contact sites (MERCS). I am personally more familiar with a similar term coined earlier (MAMs) referring to Mitochondrial associated membranes. As a recommendation, the authors may want to include a brief sentence indicating if such terms can be used for the sites of interactions of mitochondria with ER or carefully indicate their differences.
We have included the following sentence to explain how we differentiate MAMs from MERCs.
“While the regions of the ER membranes associated with mitochondria are often referred to as mitochondria associated membranes (MAMs), the term MERCS refers to the entire proteome abundant on interconnected mitochondria and ER”
L341: the calcium transfer is always from ER to mitochondria and not between them indistinctly. It must be clearly stated.
Thank you for pointing out this potential misunderstanding. We made sure to mention the directionality every time to avoid confusing the reader.
L350 “However, the release of Ca2+ ions via IP3R upon mitochondrial proximity and mitochondrial stimulation is independent of the presence of a corresponding VDAC1 molecule in the OMM” this is not clear. As the authors know, VDAC1 does not stimulate the release of calcium by the IP3R. The IP3R is a ligand activated channel. If the authors wanted to mean that ER-mitochondrial transfer via IP3 release is not dependent of VDAC1, it is truth, and they may want to discuss the presence of other VDAC isoforms.
Thank you for pointing out this mistake. We have corrected this mistake and added a paragraph discussing IP3R and VDAC isoforms.
“Notably, both, IP3R and VDAC1 are abundant in different isoforms within different cell types [130–133]. However, studies often do not discriminate between the different IP3R isoforms (IP3R1, IP3R2, and IP3R3), whereas VDAC1 is the isoform that is most studied in the context of MERCS as it is the only isoform present in co-immunoprecipitates with IP3R receptors. Therefore, this review will focus mainly on VDAC1.”
finally, all the genes and Latin terms (in vivo, in vitro, e.g., et al., ) must be in italics.
We have made the respective changes. However, according to The NCBI Style Guide [Internet]. Bethesda (MD): National Center for Biotechnology Information (US); 2004-. Chapter 5, Style Points and Conventions. Available from: https://www.ncbi.nlm.nih.gov/books/NBK995/ we opted to not italicize “e.g.” and “et al”.
Reviewer 2 Report
The manuscript sent to me for review is a systematic summary of the basic knowledge about the involvement of the contact sites between the mitochondria and the ER in the process of mitophagy. This is a specific variant of the autophagy, yet the manuscript describes not only the pathways and mechanisms of mitophagy, but also a half of the text focuses on the role of protein complexes connecting the membranes of the mitochondria and the ER, thus on the role and structure of MERCs. Too little emphasis is placed on the role of mitophagy in physiology and pathology, taking into account specific diseases. But I appreciate a detailed description of each of the protein complexes tethering the mitochondrial membrane and the ER and their roles. Unfortunately, the language in which the work is written is very unscientific and in some places the phrases are colloquial, which can confuse the reader. The lack of accuracy in terminology and precision in describing structures and biological phenomena results in the passage of mistakes. If this is due to the lack of linguistic correction, such correction must be carried out. For example, unclear sentence: “Some crosstalk between the mitochondrial 38 stress and the more established UPR of the ER exist, such as activation of activating tran-39 scription factor 4 (ATF4) signaling”.
In the first part the authors write that mitochondria play a role mostly in anabolic processes, which is not true (lines 24,25). Do the chaperones play a role only in folding of the preproteins and heat shock? What about chaperones that are inside mitochondria? Line 46 – not clear which membrane authors write about. What do the authors understand by the toxic reagents (a phrase used often in the manuscript) ex. in the line 54. An example of non-scientific expression: “While fusion decreases to prevent the 53 spread of toxic reagents across the mitochondrial network, fission increases and leads to 54 smaller mitochondria”. Line 115 “defective mitochondria 114 need to be removed while healthy mitochondria extend and fission.” not clear sentence.
I don’t understand why the paragraphs describing generally tethering proteins are in the second part of the manuscript. It would be more clear if the paper would start with the description of the MERCs, tethering complexes (or spacers – because spacers should be also mentioned, and what about their role in mitophagy) and then a specific interactions of the MERCs residing proteins would be described in mitophagy.
General comment – it should be specified that MERCs play a role in PHOSPHOLIPIDS biosynthesis and transfer not generally lipids, it would be more precise. There are other organelles involved in lipids biology – mitochondria for example in lipids metabolism, but specific circulation of phospholipids is characteristic for MERCs.
Lastly in conclusions, what do the authors mean by “list of MERCs”? line 596.
The quality of the language in which the work is written must be improved.
Author Response
We thank the reviewer for the detailed reading of our manuscript. In order to show cross talk between two topics (mitophagy and MERCS) that are not commonly reviewed together, we have intentionally chosen to first describe the two topics individually and then to perform the synthesis of the two in a separate paragraph. Therefore, readers familiar with one topic can read up on the respective other topic, skipping the part they are familiar with, and move on to the conclusions. As was mentioned by this reviewer, the “detailed description of each of the protein complexes tethering the mitochondrial membrane and the ER and their roles” was appreciated, which in our eyes support our approach to introduce them separately. Whether MERCs or mitophagy should be discussed first will be perceived differently by each reader, depending on their respective background. Coming from the mitophagy field, we have chosen to discuss the mitophagy pathways first.
We also want to thank the reviewer for the suggestion to emphasize the importance of mitophagy for physiology and especially pathophysiology, which indeed had been lacking from the manuscript. To this end we have added the following:
“Upon damage, mitochondria can enhance oxidative stress within the cell leading to cell death. Hence, maintaining a pool of healthy mitochondria is crucial for cellular survival, which is mediated by several mitochondrial quality control pathways. Defects in mitochondrial quality control and specifically mitophagy are therefore linked to the pathogenesis of several neurodegenerative disorder such as Parkinson’s disease and amyotrophic lateral sclerosis.”
“The PTEN-induced kinase 1 (PINK1)-Parkin pathway is one of the best characterized mitophagy pathways and commonly activated upon mitochondrial depolarization (Fig. 1A). Both proteins are found mutated in hereditary forms of Parkinson’s disease fitting to the general notion that mitochondrial quality control is an important determinant for neuronal health.”
Also, we are grateful that the reviewer pointed out several spots in which our choice of wording was not optimal. We have gone through the entire text and refined the unclear sentences and many more (see yellow markings in the manuscript). We hope to have sufficiently raised the quality of our expressions to pass the expectations of this reviewer.
Furthermore, we have clarified the following points:
For example, unclear sentence: “Some crosstalk between the mitochondrial 38 stress and the more established UPR of the ER exist, such as activation of activating tran-39 scription factor 4 (ATF4) signaling”.
Has been replaced by “Interestingly, both, mitochondria and ER stress response rely on activating transcription factor 4 (ATF4) signaling [11], hinting at a concerted stress response from the two interconnected organelles”
In the first part the authors write that mitochondria play a role mostly in anabolic processes, which is not true (lines 24,25).
We thank the reviewer for pointing this out. Obviously, mitochondria are also important catabolic centers, which is included in the term “metabolism”. We have altered the sentence to “Mitochondria are required for a multitude of cellular functions, including energy metabolism, calcium homeostasis, and lipid metabolism, as well as apoptotic and immune signaling”.
Do the chaperones play a role only in folding of the preproteins and heat shock? What about chaperones that are inside mitochondria?
With regards to mitochondrial quality control, the main function of chaperones lies in the support of proper protein targeting of the precursor to the organelle, as well as to support the folding once the protein is inside the organelle. We have clarified this here: “The first step of mitochondrial quality control is mediated by chaperones and proteases. Chaperones protect mitochondrial proteins from heat stress and promote correct folding of preproteins entering the organelle, both, in the cytosol as well as inside the mitochondrial matrix.”
Line 46 – not clear which membrane authors write about.
We have specified that we are talking about autophagosomal and mitophagosomal membranes.
What do the authors understand by the toxic reagents (a phrase used often in the manuscript) ex. in the line 54. An example of non-scientific expression: “While fusion decreases to prevent the 53 spread of toxic reagents across the mitochondrial network, fission increases and leads to 54 smaller mitochondria”.
We thank the reviewer for pointing out this unclear choice of words. We no longer use this term and have replaced the sentence in question it as follows: “To maintain the quality of the mitochondrial network in order to prevent spreading of ROS, defective mitochondria need to be isolated from the network and targeted for degradation.”
Line 115 “defective mitochondria 114 need to be removed while healthy mitochondria extend and fission.” not clear sentence.
We have replaced this overly colloquial term with ". To maintain the quality of the mitochondrial network to prevent spreading of ROS, defective mitochondria need to be isolated from the network and targeted for degradation."
I don’t understand why the paragraphs describing generally tethering proteins are in the second part of the manuscript. It would be more clear if the paper would start with the description of the MERCs, tethering complexes (or spacers – because spacers should be also mentioned, and what about their role in mitophagy) and then a specific interactions of the MERCs residing proteins would be described in mitophagy.
As explained above, we agree that other orders would also be suitable, but have chosen intentionally to keep the two topics separate to accommodate readers from different backgrounds. We also now mention spacer proteins in our description of MERCS, but to our knowledge, no spacer protein has been involved in mitophagic processes yet. We therefore decided to remain focussed on tethers.
“As all inter-organellar contact sites, MERCS exhibit a specific proteome and lipidome, that are required for the interplay between the organelles. Some of the proteins required to maintain inter-organellar interactions are structural proteins such as tethering proteins, which are mediating the organellar interaction (Fig. 2A), as well as spacer proteins, that define the distance between the two organelles. [122].”
General comment – it should be specified that MERCs play a role in PHOSPHOLIPIDS biosynthesis and transfer not generally lipids, it would be more precise. There are other organelles involved in lipids biology – mitochondria for example in lipids metabolism, but specific circulation of phospholipids is characteristic for MERCs.
We would like to thank the reviewer for pointing out this flaw. We now specifically mention phospholipids, instead of lipids.
Lastly in conclusions, what do the authors mean by “list of MERCs”? line 596.
This was the list of tethering protein we described in the second chapter. However, we agree that this expression was misleading and have replaced it with the following: ”To date, not all proteins mediating contact between mitochondria and ER are known, and future research will most certainly reveal other tethering proteins”